# Programmed Cell Death-Ligand 1 in Head and Neck Squamous Cell Carcinoma: Molecular Insights, Preclinical and Clinical Data, and Therapies

**DOI:** 10.3390/ijms232315384

**Published:** 2022-12-06

**Authors:** Piero Giuseppe Meliante, Christian Barbato, Federica Zoccali, Massimo Ralli, Antonio Greco, Marco de Vincentiis, Andrea Colizza, Carla Petrella, Giampiero Ferraguti, Antonio Minni, Marco Fiore

**Affiliations:** 1Department of Sense Organs DOS, Sapienza University of Rome, Viale del Policlinico 155, 00161 Rome, Italy; 2Institute of Biochemistry and Cell Biology (IBBC), National Research Council (CNR), Department of Sense Organs, Sapienza University of Rome, Viale del Policlinico 155, 00161 Rome, Italy; 3Department of Experimental Medicine, Sapienza University of Rome, 00185 Rome, Italy

**Keywords:** head and neck squamous cell carcinoma, immunotherapy, PD-1/PD-L1, immunotherapy molecular mechanism, immunotherapy resistance, pembrolizumab, nivolumab, metastatic head and neck cancer, chemotherapy

## Abstract

Aberrant expression of the programmed cell death protein ligand 1 (PD-L1) constitutes one of the main immune evasion mechanisms of cancer cells. The approval of drugs against the PD-1-PD-L1 axis has given new impetus to the chemo-therapy of many malignancies. We performed a literature review from 1992 to August 2022, summarizing evidence regarding molecular structures, physiological and pathological roles, mechanisms of PD-L1 overexpression, and immunotherapy evasion. Furthermore, we summarized the studies concerning head and neck squamous cell carcinomas (HNSCC) immunotherapy and the prospects for improving the associated outcomes, such as identifying treatment response biomarkers, new pharmacological combinations, and new molecules. PD-L1 overexpression can occur via four mechanisms: genetic modifications; inflammatory signaling; oncogenic pathways; microRNA or protein-level regulation. Four molecular mechanisms of resistance to immunotherapy have been identified: tumor cell adaptation; changes in T-cell function or proliferation; alterations of the tumor microenvironment; alternative immunological checkpoints. Immunotherapy was indeed shown to be superior to traditional chemotherapy in locally advanced/recurrent/metastatic HNSCC treatments.

## 1. Introduction

The programmed cell death protein 1 (PD-1) was isolated in 1992 [1]. It is a transmembrane protein mainly expressed by immune system cells such as T lymphocytes with a crucial role in immune self-tolerance [2,3,4,5,6]. The interaction with its ligands, the programmed cell death protein ligand 1 and 2 (PD-L1 and PD-L2), induces T-cell inhibition [2,7]. Physiologically, T-cells destroy aberrant cells, such as pathogen-infected and cancer cells, by binding the T-cell receptor (TCR) to the major histocompatibility complex (MHC) [2]. The aberrant expression of PD-L1 by cancer cells inhibits the T-cells’ cytotoxic activity, a leading mechanism of cancer immune evasion [8]. Following this discovery, new drugs active on PD-1/PD-L1 have been produced, tested, and approved [9,10,11].

Head and neck squamous cell carcinomas (HNSCC) are the sixth most common cancer worldwide [12,13,14]. The first-line regimen for recurrent or metastatic HNSCC, before the introduction of immunotherapy, was the EXTREME protocol with Cetuximab + Cisplatin or Carboplatin + 5-Fluorouracil (5-FU). The results of this protocol were not optimal. The median overall survival (mOS) was only 10.1 months, the median progression-free survival (mPFS) was 5.6 months, and the response rate (RR) was 36% [15,16]. This value dropped between 3 and 13% in the second-line treatment with a median survival of less than 6 months [16,17,18,19]. The introduction of PD-1/PD-L1 immunotherapy significantly improved this percentage, but they are still not optimal, and only 20 to 30% of patients have long-term benefits [8,20].

This literature review aims to describe the molecular mechanisms of the PD-1/PD-L1 axis underlying the aberrant expression of PD-L1 within neoplastic tissues. We also analyzed mechanisms of drug resistance and examined preclinical and clinical data on HNSCC immunotherapy. Finally, we focused on future prospects concerning new markers for patient selection, molecules, and treatment protocols.

## 2. Materials and Methods

We searched papers in the PubMed, Embase, and Cochrane Central Register of Controlled Trials databases for articles in the literature from the year of isolation of PD-1 (1992) to August 2022. Search terms included the following: PD-1 molecular mechanism, PD-L1 molecular mechanism, Pembrolizumab, Nivolumab, head and neck cancer resistance, immunotherapy, anti-PD-1, anti-PD-L1, metastatic head and neck cancer, recurrent head and neck cancer. We also hand-searched bibliographies for relevant articles. We excluded non-English language papers. All authors discussed results with conflicts solved by our senior author A.M. Hence, we have selected those papers, according to our research group, showing the most important evidence in defining the molecular mechanisms concerning PD-1-PD-L1. We have divided the evidence into paragraphs, starting from molecular structures and mechanism of action, then analyzing the mechanisms that underlie the aberrant expression in neoplasms. We then dealt with the main clinical trials concerning molecules active on PD-1 or PD-L1 in HNSCC. Finally, we examined all the possible criticalities of the present studies and the future therapeutic and diagnostic perspectives.

## 3. Results

### 3.1. Molecular Structure and Function

In 1992, Ishida Y. et al. isolated the PD-1 gene. The nucleotide sequence encodes for a 288 amino acids protein with two hydrophobic regions, the N-terminal acting as a signal peptide and the other in the middle as the transmembrane segment. After cleavage of the signal peptide, the mature form of the PD-1 protein is 268 amino acids, with an extracellular part of 147, a transmembrane of 27, and a cytoplasmic part of 94. The PD-1 extracellular domain is similar to that of the immunoglobulin superfamily. The cytoplasmatic tail is like that of most of the polypeptides associated with antigen receptors and Fc receptors (proteins found on the surface of certain cells binding to antibodies that are attached to infected cells or invading pathogens). The protein is expressed on the surface of antigen-stimulated T-cells [1]. PD-1 interacts with its ligands PD-L1 and PD-L2. Despite the PD-1/PD-L2 having a 2-6-fold higher affinity than the PD-1/PD-L1 interaction, PD-L1 is the primary inhibitor of T-cells via PD-1 binding [2,7].

#### 3.1.1. PD-1 and PD1-L1 Structures

PD-1 has a weight of about 55 kDa. It has 3 domains, an extracellular Ig-V-like domain (20 aa), a transmembrane domain, and a cytoplasmic domain with a signal transduction system with two tyrosine kinases (95 aa). The intracellular domain also contains two motifs, an immunoreceptor tyrosine-based switch motif (ITSM) and an immunoreceptor tyrosine-based inhibitory motif (ITIM). ITSMs usually deliver inhibitory signals [1,21]. The C-terminal tyrosine is related to the Src homology region 2 domain-containing phosphatase-1 (SHP-1) and SHP-2 [22,23,24,25].

#### 3.1.2. PD-1/PD-L1 Interaction

The binding between PD-1 and PD-L1 induces a PD-1 conformation change that activates the kinase cascade of the SRC family and leads to the phosphorylation of the cytoplasmic ITIM and ITSM. Which in turn activates SHP-1 and SHP-2, reducing the activation signals of T-cells. Recent studies have suggested that the main target of dephosphorylation is the CD28 co-stimulator receptor (Figure 1) [26,27,28,29,30,31,32]. PD-L1 also interacts with CD80 with an inhibitory effect on activated T cells [33].

#### 3.1.3. Biological Role

In PD-1 gene-deficient or PD-L1 genes inhibited by antibodies in animal models, it has been observed the development of autoimmune diseases and cardiomyopathies, the onset of diabetes, alteration of thymic T-cells, and impaired feta maternal tolerance [2,3,4,5,6]. Furthermore, it has been shown that PD-1 inhibition modulates the cellular T response with an increase in their responsiveness [34,35,36,37].

In T-cells, the interaction between PD-1 and PD-L1 leads to suppressive signals and dephosphorylation of the TCR pathway [38]. When PD-1 binds, its ligand causes the tyrosine phosphorylation of the cytoplasmic region and the binding of SHP-2 to the C-terminal of tyrosine in the ITSM region. Hence, SHP-2 dephosphorylates both the zeta-chain associated protein kinase 70 (ZAP70) and the CD28 PI3K pathway proteins, initiating inhibitory signaling downstream of the cascade. Furthermore, the inhibition of PI3K activation leads to reduced IL-2 production and glucose metabolism with the induction of T-cells energy (see Figure 1) [39,40,41].

It has been observed that PD-L1 does not appear to be just a ligand. Still, its binding with PD-1 seems to initiate a cascade of signals in cancer cells that induces a decreased resistance to some forms of apoptosis, reducing the activity of both mTOR and glycolytic metabolism that protects against the cytotoxic effect of interferons I and II and cytolysis mediated by cytotoxic T lymphocytes. Although it is not yet crystal, what is the intracellular signal transduction mechanism [42,43,44].

### 3.2. PD-L1 Tumor Expression

In most cases, the increase in membrane PD-L1 levels gives cancer cells an advantage in evading immune defenses. But when these subjects are treated with immunotherapy at higher levels of PD-L1, there is also a greater susceptibility to drugs [2] as observed for other types of cancer [45,46,47].

#### 3.2.1. Genetic Modifications

PD-L1 and PD-L2 are only 42 kilobases away on chromosome 9 (9p24.1). The mechanisms adopted by cancer cells to increase their transcription are both amplification and translocation (Table 1, Figure 1) [2]. Their proximity, therefore, allows a single mutation of this type to lead to an increase in the expression of both genes. Analyzing chromosome 9, Green et al. observed that Janus kinase 2 (JAK2) is located at the 9p locus. If the modification of the DNA also involves the gene that decodes for it, a further increase in the expression of PD-L1 would be obtained thanks to the action on the IFN-γ receptor pathway (Table 1, Figure 1) [48]. PD-L1 levels are also regulated by microRNAs (miRNAs) that bind the 3 ‘UTR of the gene. miRNAs play an inhibitory role at the post-transcriptional level. Its loss or modification was related to an increase in PD-L1 expression, and its deletion via CRISPR Cas9 led to greater stability of PD-L1 mRNA (Table 1, Figure 1) [2,49,50,51].

#### 3.2.2. Inflammatory Signaling

Physiologically PD-1/PD-L1 serves to regulate the action of T cells, and the discovery of PD-L1 regulation by inflammatory signaling is not surprising. Tumors, in addition to determining DNA modifications as systems to increase their expression, also exploit inflammatory molecules. IFN-γ is considered the main inducer of PD-L1 expression. The binding to the receptor activates the Janus Kinases- signal transducer and activator of transcription proteins (JAK-STAT) pathway, especially STAT1, causing the expression of interferon-responsive factors (IRFs) (Figure 1, Table 1) [54,55]. The fact that IFN-γ is not the only one responsible for PD-L1 expression is also demonstrated by the experiments of Noguchi et al., who observed how inhibition of IFN-γ by antibodies reduces but does not eliminate the expression of PD-L1 [88]. 

Lipopolysaccharide has also been observed to increase PD-L1 levels. Its mechanism of action occurs throughout the toll-like receptor (TLR) 4, which activates the nuclear factor kappa-light-chain-enhancer of activated B cells (NF-κB). The latter increases the expression of type I interferons [56,57]. TLR3 also appears to increase PD-L1 expression. The same has been observed for numerous other inflammatory factors such as IL-17, IL-10, TNF-α, IL-4, IL-1b, IL-6, and IL-27 (Table 1, Figure 1) [58,59,60,61,62].

#### 3.2.3. Oncogenic Pathways

The regulation of PD-L1 production is also regulated by some oncogenic transcription factors. MYC contributes to the tumorigenesis of 70% of all neoplasms [89]. Its pharmacological inhibition significantly reduces PD-L1 expression. The mechanism of action appears to be the link between the MYC proto-oncogene—bHLH transcription factor (MYC) and the PD-L1 promoter which causes an increase in gene transcription [63]. Consistent with this observation, MYC levels correlate with PD-L1 expression (Table 1, Figure 1) [90].

Neoplasms often overcome the obstacle of hypoxia by producing hypoxia-inducible factors (HIFs), which promote angiogenesis. HIFs also increase PD-L1 expression by the interaction between HIF-_1_A and HIF-_2_B with the hypoxia response element (HRE) on the promoter of the PD-L1 gene (Table 1, Figure 1) [64,65,66,67]. STAT3 is a molecule found downstream of many signal cascades and is known to be active in many neoplasms. It acts directly on the PD-L1 promoter and increases its expression [68,69]. In addition to activating type I interferons, as mentioned above, NF-κB has a direct action on the PD-L1 gene. This transcription factor can be activated by both oncogenic mutations and inflammatory factors. One of its subunits, p65 (or RELA), directly binds the PD-L1 promoter and increases its expression [70,71].

Another suspected molecule is serine-threonine kinase cyclin-dependent kinase 5 (CDK5), known to destabilize the competitive IRF1 repressor called IRF2 and cause the induction of PD-L1 (Figure 1, Table 1) [72]. As discussed above, the role of IFN-γ in inducing PD-L1 expression has been described by many authors. Consequently, the molecules acting on IFN-γ can also be indirectly active in this sense (Figure 1, Table 1). The AKT-mTOR cascade, located downstream of the phosphatidylinositol 3-kinase (PI3K) signaling, can be directly activated by type I and II IFN. Some authors have observed how suppression of the AKT-mTOR pathway reduces IFN-γ-induced PD-L1 expression. Moreover, the suppression of the Phosphatase and tensin homolog (PTEN) gene, which negatively regulates the signaling of PI3K-AKT, increases the expression of PD-L1 (Figure 1, Table 1) [73,74,75,76].

The increase in RAS GTPase and/or BRAF tyrosine kinase activity typical of many tumors activates the MEK-ERK pathway. Numerous studies have indicated that this pathway regulates PD-L1 expression. Again, the action is further confirmed by the observation that their inhibition reduces the transcription of PD-L1 (Figure 1, Table 1) [77,78]. K-RAS, epidermal growth factor receptor (EGFR) and anaplastic lymphoma kinase (ALK) can induce PD-L1 expression in cancer cells [91,92]. EGFR acts through mTOR and ERK-dependent mechanisms; ALK uses STAT3 and MEK-ERK (Figure 1, Table 1) [79,80,81,82].

#### 3.2.4. miRNA-Mediated Regulation

It is well known from the literature that miRNA acts as post-transcriptional regulators of gene expression targeting mRNA [2]. The expression of PD-L1 is also regulated by its action. miR-513 was the first to be identified. It binds the PD-L1 3′UTR and inhibits IFN-γ induced PD-L1 expression; IFN-γ, instead, suppresses miR-513 expression. Subsequently, many inhibitory miRNAs have been isolated, such as miR-155, -34a, 142-5p, -93, -106b, -138-5p, -217 (laryngeal cancer), -200, -152, -570, -17-5p, -15a, -193a, -16, and -197 [93,94,95,96,97,98,99,100,101,102,103,104,105,106]. By contrast, miR-20, -21, and -16 increase their expression (Table 1, Figure 1) [83].

#### 3.2.5. Protein Level Regulation

CMTM6 and 4 are the positive regulators of PD-L1 expression. The entity of the CMTM6 effect seems to vary among the tissues studied. CMTM6 binds PD-L1 and prevents ubiquitination and lysosomal degradation. CMTM6 allows PD-L1 molecules to clump together (Table 1, Figure 1) [2,84,85]. Starting from the observation that the level of PD-L1 varies during the cell cycle, Zhang et al. noted that the cascade of cyclin D-CDK4, through the phosphorylation of Speckle-type POZ protein (SPOP), elicits the ubiquitination, therefore the degradation, of PD-L1 (Figure 1, Table 1) [86]. The glycosylation of PD-L1 by glycogen synthase kinase 3b (GSK3b) increases its degradation and influences the interaction with PD-1 [107]. NF-κB, in addition to the transcriptional regulation of PD-L1, increases PD-L1 protein levels by removing ubiquitin chains via COP9 signalosome 5 (CSN5) [87].

#### 3.2.6. Mechanisms of PD-L1 Overexpression Observed in HNSCC

Some of the mechanisms listed above have also already been observed in HNSCC; however, as regards the others, this does not exclude that they can be identified in head and neck tumors with future studies. Furthermore, some mechanisms have been discovered in mouse cell models and only subsequently studied in human cells; we cannot exclude that this may also happen for HNSCC.

Gene amplification is a mechanism of PD-L1 overexpression that has been observed in HNSCC. Straub et al. observed PD-L1 expression in 45% of oral cavity carcinomas and gene amplification in 19% (with high levels in 15% and low in 4%) [52]. IFN-γ and EGFR both use JAK2 to transmit signals of extrinsic or intrinsic origin, respectively. In HNSCC, overexpression of EGFR correlated with that of JAK2 and PD-L1. Furthermore, PD-L1 expression is dependent on that of EGFR and JAK2/STAT1, and JAK2 inhibition prevents PD-L1 upregulation [53].

miRNA-217 has been observed to have a role in esophageal, ovarian, and glioma carcinoma. Subsequently, Miao et al. have also observed its action in laryngeal carcinoma, i.e., that its expression is significantly lower in neoplastic cells than in healthy ones. Insertion of miRNA-217 into cells of the Hep2 lineage reduces their ability to migrate and invade tissues, as well as their ability to proliferate while increasing apoptosis and cell necrosis. The authors, therefore, concluded its fundamental role in inhibiting metastatic cell traits and, at the same time, that its downregulation is one of the mechanisms by which laryngeal carcinoma cells become metastatic [100]. This demonstrates that many of the mechanisms observed in vivo or in other cell populations must be studied in HNSCC. 

STAT3 appears to have a very important role in regulating PD-1-PD-L1 in HNSCC. Their levels are associated, and inhibition of STAT3 downregulates that of PD-L1 [108]. A significant share of discoveries concerning the PD-1-PD-L1 axis is inherent in immune cells such as lymphocytes, dendritic cells, monocytes, macrophages, neutrophils, or even endothelial cells. These populations are not specific to a single neoplasm, so it is necessary to evaluate whether these findings can be generalized to all tumors in which PD-L1 is overexpressed. Among the inflammatory signaling molecules, IFN-γ is involved in the PD-1-PD-L1 axis in monocytes, neutrophils, dendritic cells, macrophages, and endothelial cells [109,110,111,112]. 

The involvement of INF-α and -β was also observed in the latter three cell populations. [113,114,115] TLR4 has a role in the PD-1-PD-L1 axis in macrophages, monocytes, and dendritic cells, and TLR3 in dendritic and endothelial cells [84,116,117,118,119]. Interleukins have also been widely identified as related to the action of PD-1-PD-L1, such as with IL-12 in endothelial cells or IL-27 in dendritic cells [61,113]. It has been observed that human endothelial cells produce molecules such as PD-L1 itself, capable of activating T cells. PD-L1 expression on endothelial cells is not constitutive but induced by IFN-γ and TNF-α. Furthermore, PD-L1 also has a negative feedback function on the production of these cytokines; in fact, its blockage increases their production. Furthermore, PD-L1 expression is an active regulator of T-cell-activated cytokine synthesis [109]. Even though monocytes are subject to the action of PD-L1, it has been observed that this action is regulated by NF-κB [117].

### 3.3. Immunotherapy in HNSCC

The treatment of recurrent or metastatic HNSCC before the introduction of immunotherapy had unsatisfactory results. The mOS among platinum-sensitive patients treated with the EXTREME protocol was only 10.1 months. The second line of treatment had an even significantly lower mOS of 6 months with a response rate between 3 and 13% [17]. More than 50% of advanced HNSCC had disease recurrence within 3 years. Furthermore, chemotherapy may be associated with significant toxicity and adverse effect incidence [120]. The introduction of immunotherapy gave new impetus to HNSCC treatment [9,10,11,121]. Following two Phase 3 clinical trials, two molecules have been approved for treating HNSCC, Nivolumab, and Pembrolizumab [9,10,11,20].

Nivolumab is a fully human monoclonal antibody anti-PD-1. It was approved according to a phase III trial called Checkmate-141 that compared it to single chemotherapy agents (methotrexate, docetaxel, or cetuximab) in locally advanced HNSCC that progressed within 6 months of platinum-based therapy. The nivolumab group showed better outcomes in terms of mOS, which was 7.5 months versus 5.1 months in the standard therapy group. The overall survival (hazard ratio for death, 0.70) and 1-year survival rate were longer in the Nivolumab group than with standard therapy, 36.0 vs. 16.6, respectively [11]. The Nivolumab group demonstrated better outcomes in median progression-free survival (2.0 months vs. 2.3 months), in the rate of progression-free survival at 6 months (19.7% vs. 9.9%), and in response rate (13.3 % vs. 5.8%). The 2-year survival rate was almost tripled in the Nivolumab group compared to standard therapy (16.9% vs. 6.0%), with a bigger difference in CPS PD-L1 patients. The outcomes were not influenced by the HPV status. The authors also observed a lower incidence of treatment-related adverse events in grades 3 or 4 (13.1% N 35.1%) [11].

After a short time, another molecule has shown robust results, Pembrolizumab. It is an anti-PD-1 humanized monoclonal immunoglobulin. Its interaction with PD-1 inhibits binding to its ligand PD-L1 expressed by cancer cells allowing the action of T lymphocytes against neoplastic cells. The Keynote-040 was a randomized multicentric phase III study that compared Pembrolizumab with standard therapy in patients with recurrent or metastatic HNSCC that progressed after platinum-containing treatment or patients with locally advanced disease with recurrent or progressed cancer within 3–6 months of previous multimodal therapy containing platinum. They observed an mOS in the Pembrolizumab group of 8.4 months vs. 6.9 months in the standard of the care group. The mortality at the end of the observation period was 10% lower in the Pembrolizumab group (73% vs. 83%). The incidence of treatment-related adverse events was lower in the experimental group than in the standard-of-care one (13% vs. 36%). The immunotherapy treatment also had a more durable response (18.4 vs. 5 months). The differences were more significant in the PD-L1 CPS ≥ 1 sub-population [9]. 

According to the Checkmate-141 and Keynote-040 trials, FDA and EMA approved Nivolumab and Pembrolizumab in the HNSCC treatment [20]. Pembrolizumab was then investigated as a first-line agent alone in the Keynote-048 study. This phase III multicentric clinical trial compared three different treatment protocols, the first one with Pembrolizumab alone, the second with Pembrolizumab and Platinum or 5-FU, and the third with Cetuximab and Platinum or 5-FU. The mOS in the two groups treated with Pembrolizumab was greater than the one treated with chemotherapy alone (11.5 months vs. 10.7 months) [10]. The grade 3 and 4 treatment-related adverse events occurred in 55% of patients treated with Pembrolizumab alone, 85% of the Pembrolizumab and chemotherapy group, and 83% of the Cetuximab + chemotherapy population. The percentage of deaths among the groups was 8% for Pembrolizumab alone, 12% for Pembrolizumab + chemotherapy, and 10% for Cetuximab + chemotherapy-treated patients. The response rate was higher in patients treated with the EXTREME protocol (36% vs. 19.6% in the Pembrolizumab group). However, the duration response was greater in Pembrolizumab-treated patients (22.6 months vs. 4.5 months). The authors concluded that Pembrolizumab alone is an appropriate first-line treatment for PD-L1 positive recurrent or metastatic HNSCC, and the association between Pembrolizumab and chemotherapy is an appropriate first-line treatment for recurrent or metastatic HNSCC [10].

Considering the findings in the Keynote-048 study, Pembrolizumab was approved as a first-line agent alone or in combination with Cisplatin or 5-Fu in HNSCC patients with unresectable PD-L1 combined positive score (CPS) ≥ 1 [10,20]. The CPS, or immunohistochemistry combined positive score, for PD-L1 is calculated as 100 times the number of PD-L1 positive cancer cells, lymphocytes, and macrophages, divided by the number of viable tumor cells [8]. Two other molecules, Durvalumab and Atezolizumab, have also been studied in the context of anti-PD-1-PD-L1 drugs. The results of the trials have not led to their approval in the treatment of HNSCC. 

Durvalumab is an anti-PD-L1 high-affinity IgG1. In the Hawk phase II trial, authors studied its efficacy in platinum-refractory recurrent o metastatic HNSCC with high PD-L1 expression not previously treated with immunotherapy. Durvalumab demonstrated an overall response ratio of 16.2% with better efficacy in HPV-positive cancers (overall response ratio of 29.4% in the HPV+ sub-population and 10.9% in the HPV- sub-population). The mOS was 7.1 months with particularly low toxicity demonstrated by an 8% incidence rate of treatment-related adverse events with grade ≥ 3 [122]. Durvalumab was also compared to Tremelimumab alone and associated with it. Tremelimumab is an inhibitor of cytotoxic T-lymphocyte-associated antigen 4 (CTLA-4). In the 3-arm phase II trial called Condor, the authors compared the association of Durvalumab + Tremelimumab, Durvalumab alone, and Tremelimumab in patients affected by recurrent or metastatic HNSCC with low or negative CPS for PD-L1. The association group had an mOS of 7.8%, the Durvalumab alone group of 9.2%, and the Tremelimumab alone group of 1.6%. Tremelimumab was associated with a greater incidence of adverse events in grades 3 or 4. The Tremelimumab alone group and the Durvalumab + Tremelimumab group had an incidence of treatment-related adverse events of grade ≥ 3 of 16.9% and 15.8%, respectively, compared to the Durvalumab alone group with an incidence of 12.3%. The authors concluded that there was no advantage in mOS with Durvalumab alone or in combination with Tremelimumab vs. chemotherapy. However, they observed a median 1-year and 2-year survival with Durvalumab comparable to that obtained in the Checkmate-141 trial with Nivolumab and Keynote-040 with Pembrolizumab [9,11,123]. Surely the choice made in the Condor study to have a population of patients with CPS for PD-L1 negative significantly influenced the outcome of the active drug against this axis. This makes comparing Pembrolizumab, Nivolumab, and Durvalumab trials difficult [9,11,123]. Another drug active against PD-L1 is Atezolizumab, which showed a very tolerable safety profile in a phase I trial and encouraging data about its activity with mOS of 6.0 months and median progression-free survival of 2.6 months. Its action was unrelated to PD-L1 aspiration or HPV [124].

### 3.4. Immunotherapy Resistance Mechanisms in HNSCC

Pembrolizumab and Nivolumab outperformed traditional chemotherapy in HNSCC treatment [2,10,11]. However, a significant proportion of patients do not respond to immunotherapy or develop resistance quickly. The identification of the mechanisms responsible for this phenomenon could help us to better identify patients eligible for therapy and to identify new therapeutic targets. 

The resistance mechanisms to anti-PD-1/PD-L1 immunotherapy in HNSCC were divided into four categories: tumor cell adaptation, T-cell function and proliferation, change in the tumor microenvironment, and use of alternative immune checkpoint [11]. Cancer cells respond to the selective pressure induced by immunotherapy by selecting those populations in which DNA modifications make them less sensitive to drugs. The HPV+ HNSCC cells are more prone to TRAF3 and β-2-microglobulin (β2M) mutations. The latter is part of the MHC Class I complex heavy chain, and its mutation hesitates in reducing T-cell recognition of cancer cells [125,126]. Some authors hypothesized that this mutation could be one of the mechanisms underlying the immune escape of cancer cells to anti-PD-1-PD-L1 immunotherapy [54,55]. IKZF1 is a transcription factor that induces recruitment of the immune infiltrate to tumors and increased sensitivity to PD-1 and CTLA4 inhibitors, including HNSCC. Its loss of function, on the other hand, has the opposite effect and could constitute a mechanism of resistance to immunotherapy (Table 2, Figure 2) [56,57,127]. 

We know that the action of drugs against PD-1/PD-L1 is to remove the inhibition that these molecules have against the action of T lymphocytes [11], and a possible mechanism of resistance originates from the inhibition of the T cells themselves. Cyclic GMP-AMP (cGAMP) synthase (cGAS) stimulator of interferon genes (STING) activates innate immunity against infected or neoplastic cells. This molecule is suppressed by histone H3K4 lysine demethylases KDM5B and KDM5C, and activated by H3K4 methyltransferase. In HNSCC HPV+, the values of KDM5B are inversely related to those of STING, to CXCL10 (one of the interferon-induced chemokines that promotes inflammatory infiltrate in the tumor microenvironment), and to the amount of CD8+ infiltrate. CD8+ T-cell values in the T infiltrate are directly correlated with cancer survival. Consequently, high levels of KDM5B are correlated with poor prognosis indicating this molecule is a potential target for immunotherapy (Table 2, Figure 2) [128]. The tumor microenvironment plays a fundamental role in immunotolerance. CD44+ stem-like cells are part of it in HNSCC immune microenvironment; they are capable of inhibiting T-lymphocytes proliferation and Th1 response while inducing immunosuppressive T-regulatory cells and myeloid-derived suppressor cells (MDSC) [58,59,60,61,62,129]. Their action is not dependent on the PD-1/PD-L1 pathway. Their increase in the tumor microenvironment is suspected to be an immunotherapy escape mechanism (Table 2, Figure 2) [8].

The study of the tumor microenvironment has led to the discovery of numerous molecules involved in resistance to immunotherapy with action on T-cells [130,131]. It has been observed that the increase in the expression of indoleamine 2-3-dioxygenase-1 (IDO1) reduces the proliferation not only of T-cells but also of other elements of the inflammatory infiltrate in oral squamous cell carcinoma [132]. These neoplastic cells, in addition to acting on the signals that regulate functioning and proliferation, can also act on the nutrients that the cells of the immune system need to survive. A potential mechanism observed is the increase in the expression of arginase-1 (Arg -1) by cancer cells which leads to greater degradation of L-arginine, which is a key nutrient for T and NK cells (Table 2, Figure 2) [133,134]. 

The production of TGF-β has a role in both intracellular and extracellular environments. Its secretion by cancer-associated fibroblasts inhibits CD8+ T cells and decreases the dendritic cells in draining lymph nodes [135,136,137,138]. CD8+ lymphocytes are also inhibited by CD38 via the adenosine receptor signaling and CD39 [139,140]. In addition to direct inhibition of their action, the reduction of T lymphocyte activity is also induced through the activation of polymorphonuclear myeloid-derived suppressor cells (PMN-MDSC) which, through the nitric oxide pathway, inhibits the proliferation and function of T-cells [140,141,142,143]. The activation of the nucleotide-binding domain leucine-rich repeat and of the pyrin domain containing receptor 3 (NRLP3) inflamed induced an increase in MDSCs, T-regs, and TAMs, and a reduction of IL-1β synthesis, leading to an immunosuppressive effect [144]. The action of immunotherapy escape mechanisms is also active against dendritic cells, alteration in STAT-mediated pathways, and cytokines production leading to a loss of function of dendritic cells (Table 2) [131,145,146,158].

PD-L1 is not the only immune checkpoint in HNSCC; indeed, activating alternative tolerance mechanisms leads to cancer cells’ immune escape. The lymphocyte activation gene-3 (LAG3), the pathway of the T-cell immunoglobulin and the ITIM domain, the T-cell immunoglobulin mucin-3 (TIM-3), and the V tomorrow-containing IG suppressor of T-cell activation (VISTA) are some examples of the alternative immune checkpoint that cancer cells use [8,147,148,149,150,151,152,153]. Under immunotherapy selective pressure, neoplastic cells produce all-trans retinoic acid (ATRA) and IFN-β, which increase CD38 production via the CD30-CD203a-CD73 axis. CD38 transforms NAD(+) and NADP(+) into cyclic ribose ADP (zADPR), NAADP, and ADPR, which act on calcium signaling [8,142,143,150,154,155]. Furthermore, CD73 dephosphorylates extracellular AMP which leads to the production of adenosine. Adenosine binds to the A2a and A2b receptors of T and NK lymphocytes, neutrophils, dendritic cells, and macrophages with an immunosuppressive action [139,140,141,155,156,157]. HLA, β-2-macroglobulin, and TRAF3 mutation are common in HPV+ HNSCC, whereas they are found in less than 10% of HPV cancers [126]. HPV antigens also enhance cytotoxic T-lymphocytes dysregulation (Table 2, Figure 2) [141]. Individuals with locally advanced HNSCC have a high incidence of local infections. Therefore, they are often treated with antibiotic therapy. The consequent alteration of the microbiome seems to be related to PD-1/PD-L1 drug resistance, although the mechanism by which dysbiosis leads to it is still unclear [159,160,161].

## 4. Discussion

### 4.1. Biomarkers of Immunotherapy Response in HNSCC

Although immunotherapy outcome results were encouraging, ∼60% of patients with recurrent or metastatic HNSCC do not respond to anti-PD-1/PD-L1 therapy. [8] Identifying molecular markers that, together with the CPS for PD-L1, allow us to predict the possible response to therapy early could help us in the selection of patients. As well as the identification of molecules that allow us to predict the response to treatment during the course of the same could help us manage therapies. IFN-γ upregulates PD-L1 and PD-L2 to downregulate the cytotoxic response. IFN-γ active signaling is associated with anti-PD-L1 therapy response, and IFN-γ-related mRNA profile predicts clinical response to PD-1 blockade in HNSCC [162].

Hypoxia within the tumor microenvironment leads to an “immune desert”, a decrease in immune cells that is a further immune evasion mechanism. The resulting paucity of T-cells explains the poor response to PD-1/PD-L1 immunotherapy. The biomarkers that can highlight this phenomenon in HNSCC are hypoxia-inducible factor-1α (HIF-1α) and its signaling [64,163,164]. The tumor microenvironment and the cells contained in it might have a role in immunotherapy resistance. Intriguingly, head and neck cancer-associated fibroblasts (HNCAF) modulate the immune response to HNSCC and could be used as potential immunotherapy response biomarkers [165].

Usually, the higher the number of DNA mutations, the more neoantigens are presented to the antigen-presenting cells (APCs). The more the cancer cells are susceptible to cytotoxic killing by T-lymphocytes. A defect in DNA mismatch repair genes (hMLH1 and hMSH2) causes the accumulation of DNA mutations and microsatellite instability. It leads to a high tumor mutation burden (TMB-high). It has been observed that higher TMB predicted response to anti-PD-1/PD-L1 in head and neck cancers [166]. The tumor cell mutation burden is also a predictor of patient survival in HNSCC when measured as “peripheral blood tumor cell mutation burden” (bTMB). bTMB, TMB, and inflammatory biomarkers are considered independent predictors of Pembrolizumab efficacy [167,168].

### 4.2. Future Perspectives in HNSCC

Despite the improvement that immunotherapy has brought to treating HNSCC, there is a significant percentage of patients who have no long-term benefit. Currently, we perform the patient selection by evaluating the PD-L1 expression only. A wider marker-based patient selection, also based on other molecules, could help define the best therapeutic approach for each patient [20]. In this way, a personalized treatment protocol could be identified for each individual’s oncological profile. Indeed, PD-L1 levels have a predictive value of the response to immunotherapy. In some patients, although the CPS for PD-L1 is greater than 1, there is no response to therapy. Seiwert et al. proposed measuring, together with the expression of PD-L1, the levels of IFN-γ in HNSCC, as they directly influence the expression of PD-L1, and, likewise, to ensure that PD-L1 expression is related to T cell activity and not inflammation of the tumor microenvironment [169].

Numerous additional factors seem to influence the response to PD-1/PD-L1 immunotherapy. One of these is the amount of non-synonymous DNA mutations whose increase causes a greater presence of neoantigens with a consequent greater cellular T response. Tumors with a higher rate of these mutations appear to have a greater susceptibility to PD-1 and PD-L1 inhibitors [170,171].

Another advanced hypothesis was to use CMTM6 as a target, together with PD-L1, to reduce the expression of PD-L1 induced by IFN-γ [2]. Despite the enthusiasm regarding Pembrolizumab and Nivolumab, most HNSCC patients will not have long-term benefits from immunotherapy treatment. Several trials test drug associations with multiple immunotherapies or a combination of them with traditional chemotherapy [172]. The rationale behind the association of immunotherapy and traditional chemotherapy is the observation that the latter makes cancer cells more recognizable by the immune system. By combining them with the anti-PD-1/PD-L1 drugs, we obtain the combination of two treatments. One that makes cells more visible to the immune system and one that takes away the inhibition of T-cells. This effect was also observed with lower doses of cytotoxic drugs (such as Cisplatin). Reducing the amount of drug administered induces fewer adverse events, especially bone marrow hematopoiesis inhibition [173,174,175]. The TPextreme trial studied the association between taxane chemotherapy followed by a second-line treatment with immunotherapy in R/M HNSCC. The observed mOS was 21.9 months. These results have not been compared with those of immunotherapy alone [176]. 

Another possible target that appears to be correlated with surviving immunotherapy is KDM5B. It regulates the levels of STING, CXCL10, and therefore inflammatory infiltrates of the tumor microenvironment, in particular CD8+ T lymphocytes, which in turn correlate with survival. For this reason, some authors have indicated KDM5B as a potential target for immunotherapy in HNSCC [128].

Obviously, not all associations have shown results superior to the treatments already approved. According to the Eagle trial, it was observed that there was no synergy between the two molecules in R/M HNSCC; probably because Tremelimumab is an IgG2 that does not induce cell death via an antibody-dependent mechanism, while NK cells are the most numerous lymphocytes in the HNSCC tumor microenvironment [17]. In any case, the combination of anti-PD-L1 and anti-CTLA4 drugs is still under study. We are waiting for the Checkmate-651, Checkmate-714, and Kestrel trials. The former compares the association between Nivolumab (anti-PD-1) and Ipilimumab (anti-CTLA4) vs. standard therapy (EXTREME protocol). Checkmate-714 compares the same drug association (Nivolumab and Ipilimumab) with Nivolumab alone in platinum-sensitive and platinum-resistant diseases. The Kestrel trial is a tree-arm study in HNSCC platinum-sensitive patients that evaluates the association between Tremelimumab (anti-CTLA4) and Durvalumab (anti-PD-L1) vs. Durvalumab alone vs. EXTREME protocol. Obviously, anti-CTLA4 molecules are not the only ones to be studied in association with anti-PD-L1 drugs. The vascular endothelial growth factor (VEGF) is also an immunosuppressive molecule. The association between its inhibitors and anti-PD-1 therapies is under examination in the LEAP-010 trial. The results seem encouraging, with a superior anti-tumor activity compared with single molecules [177,178,179]. 

### 4.3. New Molecules in HNSCC

Several new molecules are currently in various stages of study. Monalizumab is an anti-NKG2A receptor humanized antibody. This receptor is expressed on CD8+ and NK lymphocytes. The Upstream trial is currently evaluating its efficacy alone and in combination with Durvalumab vs. standard care protocols in R/M HNSCC [180]. The Interlink-1 study analyzes the outcome of the association between Monalizumab and Cetuximab [181]. GSK609 is an anti-T-cell inducible co-stimulatory receptor (ICOS) monoclonal antibody. ICOS is involved in T-cells proliferation, differentiation, and survival. We are evaluating the results of a phase III trial INDUCE-3 which compares the association of Pembrolizumab, Platinum/5-FU, and GSK609 vs. Pembrolizumab and Platinum/5-FU [182].

### 4.4. HNSCC Therapy

The mOS observed with Pembrolizumab-based protocols is greater than standard chemotherapy in patients with recurrent or metastatic HNSCC. At higher PD-L1, CPS values correspond to a major treatment response [10]. Immunotherapy also has a better safety profile, with 2.7 times less incidence of treatment-related adverse events. Furthermore, it has greater durability [9]. The two approved drugs in HNSCC treatment, Pembrolizumab and Nivolumab, had almost equal 1-year survival rates (37% and 36%). However, this result is achieved by applying protocols with very different doses. The Pembrolizumab posology was 75 mg/m^2^ every 3 weeks in the Keynote-040 trial, and the Nivolumab one was 30–40 mg/week in the Checkmate-141 trial. Furthermore, the populations of the two trials had slightly different eligibility criteria. The Keynote-040 considered patients with disease progression between 3 and 6 months, and the Checkmate-141 patients with disease progressed within 6 months of platinum-based therapy. Hence the population in this study was also composed of patients that never responded to therapy, usually considered with a worse prognosis [9,11]. 

The populations of Checkmate-141 and Keynote-040 trials also differed in PD-L1 CPS values. In Keynote-040, over 75% of patients had CPS for PD-L1 ≥ 1; in Checkmate-141, it was 72%. This small difference could have influenced outcomes because it is well-known from the literature that a value greater than this threshold significantly influences the immunotherapy response [9,10,11]. In the Eagle trial, there was an over-performance of the standard of care group compared to the Pembrolizumab and Nivolumab trials. There was also an over-mortality in the initial period of the immunotherapy administration [17]. The over-mortality and the high rate of progressive disease have also been observed in Keynote-048 trials. The Pembrolizumab alone survival curve was below chemotherapy one over the first eight months. Then they crossed with the stabilization of a better outcome for immunotherapy-treated patients. This high rate of progressive disease at the beginning of the treatment could be explained by the “hyper-progression” phenomenon, a faster growth of cancer cells after immunotherapy initiation [17]. The excess of early deaths in Pembrolizumab alone patients was eliminated in the Keynote-048 study with the addition of chemotherapy to it. However, it comes at a price of higher toxicity. This association has a significant advantage only in CPS PD-L1 ≥ 1 compared to chemotherapy only [10,17]. 

The findings of the Keynote-048 trial led to the approval of Pembrolizumab alone for CPS ≥ 1 patients, and in association with chemotherapy for any CPS. The EMA approved it alone or in association only with CPS ≥ 1 patients. It has been observed that early Pembrolizumab-based therapy in PD-L1 CPS ≥ 1 may sensitize the tumor to subsequent therapy thanks to microenvironment modification. The outcome of therapy given after immune checkpoint inhibition is greater than predicted historically in patients who both respond and do not respond to checkpoint inhibition [183,184,185,186,187,188]. Anti-PD-L1 therapy has higher efficacy in male and smoker patients. A possible explanation for this finding is that smoking induces greater immunogenicity by increasing genetic mutations. It makes neoplastic cells more recognizable by T cells whose action is not stopped for the immunotherapy inhibition of PD-L1 [189,190,191,192,193,194,195]. Clinical staging of HNSCC plays an important role in immunotherapy efficacy. Indeed, Botticelli et al. observed that anti-PD-1 is more effective in metastatic diseases and anti-PD-L1 in recurrent ones. The possible explanation for this finding is the systemic effect given by PD-1 inhibition that strikes every circulating lymphocyte. The metastases can be attacked by active T-cells that are no longer stoppable by cancer PD-L1. The anti-PD-L1 drugs showed greater local efficacy in cancer with lower heterogeneity [20].

### 4.5. Adverse Events in HNSCC Immunotherapy

The most common side effect of anti-PD-L1 agents is autoimmune endocrinopathies [11]. Particularly, the most commons were fatigue and hypothyroidism when associated with Pembrolizumab alone therapy. The association with chemotherapy and the association of Cetuximab and chemotherapy had a higher incidence of anemia and nausea [10]. Bleeding is sometimes a side effect of immunotherapy. The bleeding incidence for Pembrolizumab and the association of Pembrolizumab and chemotherapy were 7% and 9%, respectively; the Cetuximab and chemotherapy group had a bleeding incidence of only 5% [10].

## 5. Conclusions

The introduction of anti-PD-L1 immunotherapy has significantly improved cancer pharmacotherapy. But the way to achieve satisfactory results is still impervious. The prospects regarding the new molecules, the new association protocols, and the new biomarkers are encouraging. Thanks to them, it is possible to move in the direction of personalized medicine, in which the choice of therapy is not based on the patient’s pathology but on the molecular characteristics of each patient. To date, immunotherapy constitutes a fundamental weapon in the oncology of the cervical and facial area and beyond. 

## Figures and Tables

**Figure 1 ijms-23-15384-f001:**
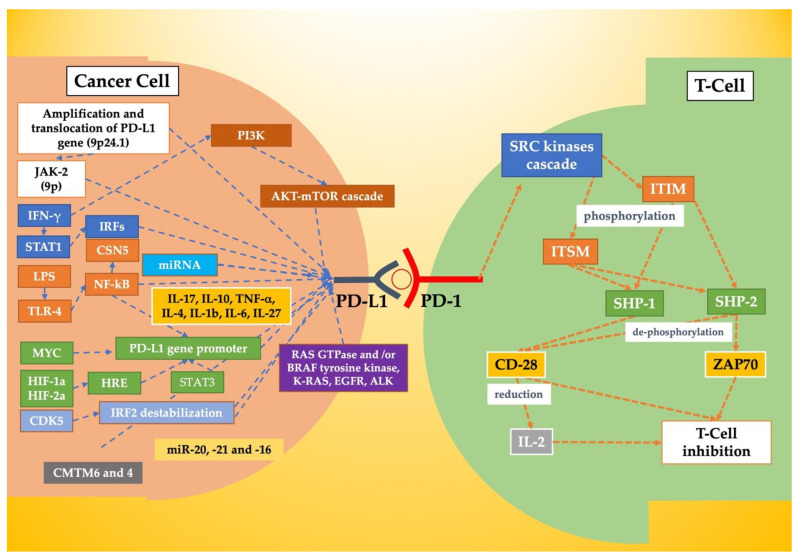
PD-1/PD-L1 mechanism of action and PD-L1 expression in cancer cells. ITIM, immunoreceptor tyrosine-based inhibitory motif; ITSM, immunoreceptor tyrosine-based switch motif; SHP-1, SHP-2, tyrosine phosphatases; ZAP70, zeta-chain associated protein kinase 70; IL-2, interleukin-2; JAK-2, Janus kinase 2; IFN-γ, interferon γ; IRFs, interferon responsive factors; LPS, lipopolysaccharide; TLR-4, toll-like receptor 4; NF-κB, nuclear factor kappa-light-chain-enhancer of activated B cells; HIF, hypoxia-inducible factors; HRE, hypoxia responsible elements; CDK5, kinase cyclin-dependent kinase 5; EGFR, epidermal growth factor receptor; ALK, anaplastic lymphoma kinase; miRNA, micro-RNA; CSN5-COP9, signalosome complex subunit 5.

**Figure 2 ijms-23-15384-f002:**
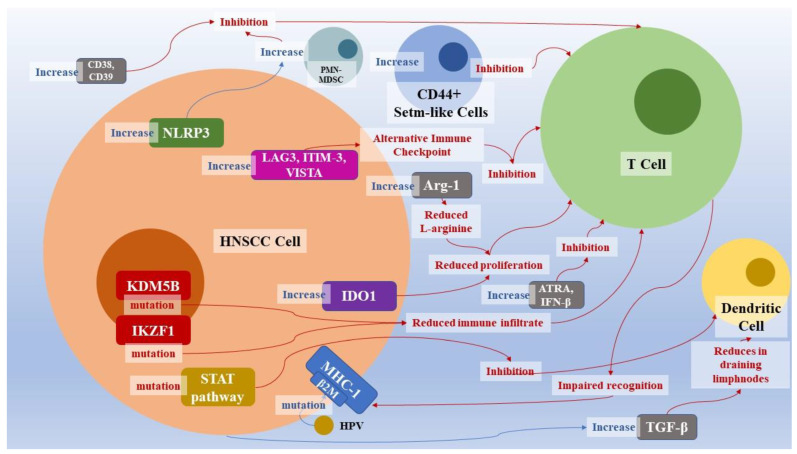
Immunotherapy resistance mechanisms in HNSCC.

**Table 1 ijms-23-15384-t001:** PD-L1 tumor hyperexpression mechanisms.

Mechanism/Molecule	Mechanism of Action	Refs
**Genetic modifications**		
Amplification	Amplification and translocation of PD-L1 gene (9p24.1)	[2,52]
Translocation		
JAK-2	JAK-2 hyperexpression due to amplifications and/or translocation (9p)	[48,53]
miRNAs	3′UTR binding	[2,49]
**Inflammatory signaling**		
IFN-γ	IFN-γ activates JAK-STAT pathway (especially STAT1), causing the expression of IRFs	[53,54,55]
LPS	LPS activates TLR4, which activates NF-κB. The latter increases type I interferons expression	[56,57]
IL-17, IL-10, TNF-α, IL-4, IL-1b, IL-6, IL-27	Inflammatory factors that enhance PD-L1 expression.	[58,59,60,61,62]
**Oncogenic pathways**		
MYC	MYC interacts with the PD-L1 promoter which causes an increase in gene transcription	[63]
HIF-1a, HIF-2a	HIF-1a and HIF-2a interact with HRE and activate the PD-L1 promoter	[64,65,66,67]
STAT3	STAT3 acts on PD-L1 promoter increasing his expression	[68,69]
NF-κB	NF-κB p65 (RELA) subunit binds PD-L1 promoter and increases his expression	[70,71]
CDK5	CDK5 destabilizes IRF2-inducing PD-L1 expression	[72]
AKT-mTOR cascade	PI3K activates AKT-mTOR cascade, that cascade increases PD-L1 expression	[73,74,75,76]
RAS GTPase and/or BRAF tyrosine kinase activity	Increase in PD-L1 expression	[77,78]
K-RAS, EGFR, ALK	K-RAS induces PD-L1 expression. EGFR acts through mTOR and ERK-dependent mechanisms. ALK uses STAT3 and MEK-ERK.	[79,80,81,82]
**miRNA-mediated regulation**		
miR-513, -155, -34a, 142-5p, -93, -106b, -138-5p, -217 (laryngeal cancer), -200, -152, -570, -17-5p, -15a, -193a, -16 and -197	PD-L1 suppressors	[79,80,81,82,83,84,85,86,87,88,89,90,91,92]
miR-20, -21 and -16	PD-L1 enhancers	[83]
**Protein level regulation**		
CMTM6 and 4	CMTM6 and 4 bind PD-L1 and prevent ubiquitination and lysosomal degradation.	[2,84,85]
D-CDK4 (loss of function)	D-CDK4 phosphorylates SPOP and elicits the ubiquitination and degradation of PD-L1	[86]
NF-κB	NF-κB removes ubiquitin chains via CSN5	[87]

**Table 2 ijms-23-15384-t002:** Immunotherapy resistance mechanisms in HNSCC.

Molecule	Mechanism of Action	References
β2M mutations	Component of the MHC Class I heavy chain, his mutation hesitates in reduction of T cell recognizant of cancer cells	[54,55]
IKZF1	Mutation that reduces the inflammatory infiltrate	[56,57]
KDM5B	Suppresses STING, CXCL10 levels, CD8+ infiltrate	[128]
Increase of CD44+ stem-like cells	CD44+ stem-like cells inhibit T-cells and enhance immunosuppressive T-reg cells	[8]
CD69 sufficient state	T-cells exhaustion	[130,131]
GCP1	Inhibition causes T-cells maturation prevention	
BH4	Reduction of BH4 inhibited by kineurine T-cells inhibiting	
IDO1	Increase of IDO1 reduces T-cells and inflammatory cells proliferation	[132]
Arg-1	Arg-1 increase expression leads to greater degradation of L-arginine, a key nutrient for lymphocytes	[133,134]
TGF-β	Decrease dendritic cells in drainage lymph nodes and CD8+ cells.	[135,136,137,138]
CD38, CD39	CD8+ cells inhibition via adenosine receptor	[139,140]
PMN-MDSC	PMN-MDSC activates the nitric oxide pathway which inhibits the proliferation and function of T-cells	[140,141,142,143]
NRLP3	NRLP3 activation increases MDSCs, T-regs, and TAMs, and reduces IL-1 β	[144]
STAT-pathway and cytokines	Alteration of STAT-pathway leads to a dendritic cell loss of function	[131,145,146]
LAG3, pathway of T-cell immunoglobulin, ITIM domain, TIM-3, VISTA	Alternative immune checkpoint	[8,147,148,149,150,151,152,153]
ATRA and IFB-β	ATRA and IFB-β increase CD38 production via CD30-CD203a-CD73 axis. CD38 transforms NAD^+^ and NADP^+^ into cyclic ribose ADP (zADPR), NAADP, and ADPR, which act on calcium signaling	[8,142,143,150,154,155]
CD73	dephosphorylates extracellular AMP which leads to the production of adenosine. Adenosine binds to the A2a and A2b receptors of T and NK lymphocytes, neutrophils, dendritic cells, and macrophages with an immunosuppressive action	[139,140,141,155,156,157]
HLA, β-2-macroglobulin and TRAF3 mutation	common in HPV+ HNSCC, whereas they are found in less than 10% of HPV− cancers. HPV antigens also enhance cytotoxic T-lymphocytes dysregulation.	[126]

## Data Availability

Not applicable.

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
