# Peer review of "Programmed Cell Death-Ligand 1 in Head and Neck Squamous Cell Carcinoma: Molecular Insights, Preclinical and Clinical Data, and Therapies"

_ijms, 2022, doi:10.3390/ijms232315384_

Round 1
Reviewer 1 Report
In this review, association of PD1/PDL1 pathway in the development of head and neck squamous cell carcinoma (HNSCC) was discussed. Following comments can be made:
1. 1. The contents of the review should be stated first followed by Abstract of the review.
2. 2. In Abstract, Objective should be focused and outcome of the review should be stated clearly.
3. 3. In sections 3.1 and 3.2, generalized description of PD1/PDL1 has been made without focusing on their alterations in HNSCC. Figure and Table should be based on the involvement of PD1/PDL1 in HNSCC along with its immune suppression mechanism.
4. 4. In sections 3.3 and 3.4, therapeutic implications of PD1/PDL1 should be discussed in reference to HNSCC only. Discussion should be made through diagram/Table for better understanding and should also be concise.
5. 5. In section 4, Future perspectives of PD1/PDL1 in therapy of HNSCC should be discussed. The discussion should be mechanistic and concise.
Author Response
Answers to the criticisms raised by reviewer 1
We did appreciate the comments of the reviewer aiming to improve the scientific quality of the paper.
In this review, association of PD1/PDL1 pathway in the development of head and neck squamous cell carcinoma (HNSCC) was discussed. Following comments can be made:
- The contents of the review should be stated first followed by Abstract of the review.
Reply: according to the comment of the reviewer, we rewrote the abstract including the contents of the review (lines 19-34 of the revised paper, test highlighted in light yellow)
- In Abstract, Objective should be focused and outcome of the review should be stated clearly.
Reply: according to the comment of the reviewer, we rewrote the abstract according also to the editorial instructions of IJNS (lines 19-34 of the revised paper, test highlighted in light-yellow).
- In sections 3.1 and 3.2, generalized description of PD1/PDL1 has been made without focusing on their alterations in HNSCC. Figure and Table should be based on the involvement of PD1/PDL1 in HNSCC along with its immune suppression mechanism.
Reply: we do thank the reviewer for pointing this out to us. To better focus the discussion on HNSCC, we have inserted a new section - 3.2.6 - dealing with the mechanisms concerning head and neck cancers (lines 232-274 of the revised paper, test highlighted in light-yellow). We did also update the Tables and include an additional Figure (2) to better explain this crucial issue.
- In sections 3.3 and 3.4, therapeutic implications of PD1/PDL1 should be discussed in reference to HNSCC only. Discussion should be made through diagram/Table for better understanding and should also be concise.
Reply: as requested, Section 3.4 was revised focussing only on HNSCC. We have made the corresponding corrections to Table 2. To reflect these changes, we have changed the title of section 3.4 to "Immunotherapy resistance mechanisms in HNSCC". Figure 2 was inserted in section 3.4 too. Furthermore, we have ensured that section 3.3 contains only trials focusing on immunotherapy of HNSCC (section 3.4 of the revised paper, text highlighted in light-yellow).
- In section 4, Future perspectives of PD1/PDL1 in therapy of HNSCC should be discussed. The discussion should be mechanistic and concise.
Reply: as suggested, section 4 was revised to better explain putative future perspectives of PD1/PDL1 in the therapy of HNSCC. To reflect these changes, we have changed the titles of subsections 4.1, 4.2 and 4.3 (section 4 of the revised paper, text highlighted in light-yellow).
Reviewer 2 Report
The review paper covers extensive background and history of how PD-1 was discovered and used as a cancer immunotherapeutic agent. The authors also discussed the potential culprits of why only a small subset of patients gained clinical benefits from anti-PD1/L1 antibody treatment. Generally, it's a comprehensive review and well-written review article. I have only one suggestion: the authors described some potential mechanisms against the therapeutic effects of anti-PD1/PDL1 treatment. It would be great if these can be illustrated in a schematic to help the readers better understand the contexts.
Author Response
Answers to the criticisms raised by reviewer 2
The review paper covers extensive background and history of how PD-1 was discovered and used as a cancer immunotherapeutic agent. The authors also discussed the potential culprits of why only a small subset of patients gained clinical benefits from anti-PD1/L1 antibody treatment. Generally, it's a comprehensive review and well-written review article.
Reply: we do thank the reviewer for the positive comments
I have only one suggestion: the authors described some potential mechanisms against the therapeutic effects of anti-PD1/PDL1 treatment. It would be great if these can be illustrated in a schematic to help the readers better understand the contexts.
Reply: as suggested to better schematize the mechanisms of drug resistance, we have included a new figure, Figure 2 inserted in section 3.4. We have also improved the text of the discussion to make such mechanisms clearer (text highlighted in light yellow throughout the revised paper).
Reviewer 3 Report
In this review, association of PD1/PDL1 pathway in the development of head and neck squamous cell carcinoma (HNSCC) was discussed. Following comments can be made:
1. 1. The contents of the review should be stated first followed by Abstract of the review.
2. 2. In Abstract, Objective should be focused and outcome of the review should be stated clearly.
3. 3. In sections 3.1 and 3.2, generalized description of PD1/PDL1 has been made without focusing on their alterations in HNSCC. Figure and Table should be based on the involvement of PD1/PDL1 in HNSCC along with its immune suppression mechanism.
4. 4. In sections 3.3 and 3.4, therapeutic implications of PD1/PDL1 should be discussed in reference to HNSCC only. Discussion should be made through diagram/Table for better understanding and should also be concise.
5. 5. In section 4, Future perspectives of PD1/PDL1 in therapy of HNSCC should be discussed. The discussion should be mechanistic and concise.
Author Response

(The authors gave the same response as above.)
